# Regional genetic correlations highlight relationships between neurodegenerative disease loci and the immune system

Frida Lona-Durazo[1,2], Regina H. Reynolds [3,4], Sonja W. Scholz[5,6], Mina Ryten [3,4,7] & Sarah A. Gagliano Taliun [1,8✉]

Neurodegenerative diseases, including Alzheimer's and Parkinson's disease, are devastating complex diseases resulting in physical and psychological burdens on patients and their families. There have been important efforts to understand their genetic basis leading to the identification of disease risk-associated loci involved in several molecular mechanisms, including immune-related pathways. Regional, in contrast to genome-wide, genetic correlations between pairs of immune and neurodegenerative traits have not been comprehensively explored, but could uncover additional immune-mediated risk-associated loci. Here, we systematically assess the role of the immune system in five neurodegenerative diseases by estimating regional genetic correlations between these diseases and immune-cell-derived single-cell expression quantitative trait loci (sc-eQTLs). We also investigate correlations between diseases and protein levels. We observe significant (FDR < 0.01) correlations between sc-eQTLs and neurodegenerative diseases across 151 unique genes, spanning both the innate and adaptive immune systems, across most diseases tested. With Parkinson's, for instance, *RAB7L1* in CD4+ naïve T cells is positively correlated and *KANSL1-AS1* is negatively correlated across all adaptive immune cell types. Follow-up colocalization highlight candidate causal risk genes. The outcomes of this study will improve our understanding of the immune component of neurodegeneration, which can warrant repurposing of existing immunotherapies to slow disease progression.

[1] Montréal Heart Institute, Montréal, QC, Canada. [2] Université de Montréal, Montréal, QC, Canada. [3] Genetics and Genomic Medicine, Great Ormond Street Institute of Child Health, University College London, London, UK. [4] Aligning Science Across Parkinson's (ASAP) Collaborative Research Network, Chevy Chase, MD, USA. [5] Neurodegenerative Diseases Research Unit, National Institute of Neurological Disorders and Stroke, Bethesda, MD, USA. [6] Department of Neurology, Johns Hopkins University Medical Center, Baltimore, MD, USA. [7] NIHR Great Ormond Street Hospital Biomedical Research Centre, University College London, London, UK. [8] Department of Medicine & Department of Neurosciences, Université de Montréal, Montréal, QC, Canada. ✉email: sarah.gagliano-taliun@umontreal.ca

Adult-onset neurodegenerative diseases, such as Alzheimer's disease (AD), Parkinson's disease (PD), and amyotrophic lateral sclerosis (ALS), are devastating conditions affecting populations worldwide and resulting in a large physical and psychological burden to patients and their families. Neurodegeneration occurs when the cells in the nervous system deteriorate, leading to cell death caused by various pathological factors and possibly multiple biological systems. Large-scale genome-wide association studies (GWASes), which have mostly been conducted in individuals of European genetic ancestry, identified multiple risk loci associated with AD, PD, ALS, and other complex neurodegenerative diseases[1–7]. However, there is a gap in our understanding of the mechanisms by which genetic risk factors influence the pathogenesis of neurodegenerative diseases.

Computational and experimental evidence of several complex neurodegenerative diseases suggest that the immune system is involved in disease development[4,7–11] (Supplementary Fig. 1). For instance, there is experimental evidence in transgenic mice suggesting an association between AD progression and accumulation of B cells and immunoglobulin deposits around Aβ plaques[9]. In addition, computational evidence has shown a genetic enrichment of AD GWAS signals across the innate and adaptive immune systems[8,9]. Further, an experimental study uncovered the role of CD4+ T cells in brains of Lewy Body Dementia (LBD) patients and its relationship with neurodegeneration[12]. In the case of ALS, the largest GWAS identified the Human Leukocyte Antigen (HLA) region as a novel disease risk locus, and an epigenome-wide association study highlighted an enrichment of Immunoglobulin E as associated with disease risk[5,13]. In addition, the immune system is thought to be implicated in PD, in which the gene *LRRK2* is a shared disease risk for Crohn's disease (CD) and PD[10,14]. Further, tyrosine kinase inhibition has shown to modulate the immune response in PD[15]. These findings motivate the study of links between neurodegeneration and immune processes. The wealth of large-scale omic data becoming available presents a unique opportunity to apply new data-driven approaches to better understand the molecular and cellular immune-related mechanisms influencing neurodegenerative diseases, through the lens of genetics.

Pinpointing targets for neurodegenerative diseases in specific cell types involved in the immune system will be key to downstream repurposing of existing immune therapies as treatment options for certain neurodegenerative diseases. Here, we identify and investigate shared genomic loci between immune function and risk of neurogenerative disease risk using bioinformatics tools with large-scale GWAS datasets and expression quantitative trait loci (gene expression and protein). In our investigation of the role of immune-mediated pathways in neurodegenerative diseases, we not only demonstrate known relationships among genes, cell types and diseases, but also identify, to the best of our knowledge, new potential links. Our approach pinpoints pertinent genes in a particular cell type for a particular neurodegenerative disease.

## Results

### Genome-wide overview of genetic correlations among diseases.
We performed pairwise genome-wide genetic correlations ($r_g$) across GWAS datasets[16]. These GWAS datasets include five neurodegenerative diseases (AD, PD, LBD, ALS and FTD) immune-mediated diseases (MS, UC and CD) and SCZ, a neuropsychiatric disorder. Using a Bonferroni-corrected $p$-value = 0.0014, we identified six significant positive correlations (Fig. 1), of which one was between two neurodegenerative diseases: PD and LBD ($r_g = 0.65$; $p$-value = 1e−03). We did, however, observe nominally significant correlations ($p$-value < 0.05) between other

pairs of tested neurodegenerative diseases, except for FTD, for which there were no nominally significant correlations (Supplementary Table 1). Between immune-mediated diseases (our control traits), we confirmed the expected significant positive correlations across these diseases (Fig. 1). Finally, we saw positive significant correlations between SCZ and immune-mediated diseases (i.e. UC and CD), but not between SCZ and neurodegenerative diseases (Fig. 1). All genome-wide genetic correlations results are provided in Supplementary Table 1.

### Regional genetic correlations highlight pleiotropic loci implicated in neurodegenerative diseases.
Reassured by the detection of known global genetic correlations and cognisant of the fact that regional correlations between two traits can be masked when assessing in a genome-wide basis[17], we estimated regional genetic correlations using LAVA[18]. The advantage of this tool is that it can perform correlations across multiple traits simultaneously. We performed a total of 1,902 pair-wise correlations across 389 loci with adequate univariate signal, yielding a total of 59 genomic regions (i.e. LD blocks—See Methods) with significant correlations in at least one trait pair (Bonferroni-corrected $p$-value threshold = 2.629e−05).

We identified significant regional correlations between various diseases and genomic loci, including loci that contain genes known to be implicated in neurodegenerative diseases. For example, we observed positive genetic correlations between AD and LBD at two genomic loci. The locus located on chromosome 2 [chr2:126754028-127895644] contains the *BIN1* gene ($r_g = 0.564$; $p$-value = 9.80e−06), whereas the locus on chromosome 19 [chr19:45040933-45893307] contains the *APOE* gene ($r_g = 0.80$; $p$-value = 1.97e−124). Both genes have been implicated in AD and LBD risk[1,2,4,19,20]. We also observed a positive genetic correlation between PD and LBD at a locus on chromosome 4 [chr4:812416-1529267] containing *TMEM175* ($r_g = 0.648$; $p$-value = 1.49e−05). In contrast, other genomic loci containing genes that are known to be involved in more than one neurodegenerative disease did not yield significant correlations, such as the locus containing *SNCA* [chr4:90236972-91309863] between LBD and PD ($r_g = 0.165$; p-value = 0.130), and the locus that includes *GRN* [chr17:42348004-43460500], known to be involved in AD, PD, FTD and LBD. In the latter case of *GRN*, AD, FTD and LBD did not have sufficient univariate signal ($p$-value ≥ 2.63e−05) to test genetic correlations at that locus. In the case of the locus including *SNCA*, the lack of correlation may be explained by previous colocalization analyses, which have suggested that there are different regulatory causal variants implicated in PD and LBD[19].

Aside from observing significant correlations at known pleiotropic loci, we also saw significant genetic correlations between neurodegenerative and immune-mediated diseases. One such example was a genomic locus on chromosome 1 [chr1:161054077-161945442] containing, among multiple genes, *FCGR2A*. This locus was negatively correlated between PD and UC ($r_g = −0.652$; $p$-value = 1.64e−05), positively correlated between UC and CD ($r_g = 0.835$; $p$-value = 1.81e−06), and it was correlated with nominal significance between PD and CD ($r_g = −0.462$; $p$-value = 0.018) (Fig. 2; Supplementary Fig. 2; Supplementary Data 1). The gene *FCGR2A* has been previously associated with several immune-mediated diseases, and it is a risk locus for PD[7,21]. In addition, a locus [chr4:169555115-170682809] that includes six protein-coding genes (i.e. *PALLD, CBR4, NEK1, CLCN3, C4orf27* and *SH3RF1*) was significantly correlated between PD and UC ($r_g = 0.525$; $p$-value = 1.02e−05) and nominally significant between ALS and CD ($r_g = 0.369$; $p$-value = 0.044) (Fig. 2; Supplementary Fig. 3). The gene *CLCN3*

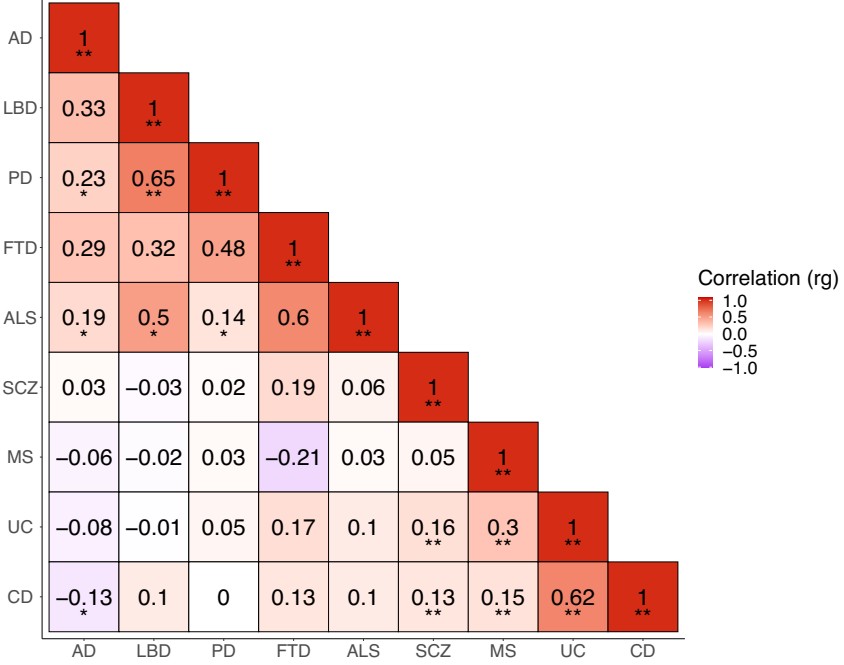

**Fig. 1 Genome-wide genetic correlations ($r_g$) across GWAS traits.** Positive correlations are highlighted in shades of red, whereas negative correlations are highlighted in shades of purple. GWAS sample sizes are presented in Table 1. ** = significant correlations ($p$-value < 0.0014); * = nominally significant correlations ($p$ < 0.05). AD Alzheimer's disease, LBD Lewy body dementia, PD Parkinson's disease, FTD Frontotemporal dementia, ALS Amyotrophic lateral sclerosis, SCZ Schizophrenia, MS Multiple sclerosis, UC Ulcerative colitis, CD Crohn's disease.

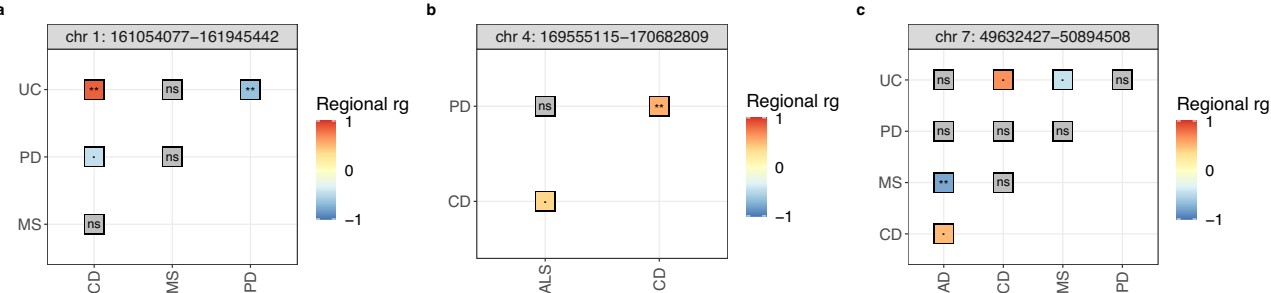

**Fig. 2 Significant regional genetic correlations ($r_g$) between neurodegenerative and immune-mediated diseases.** Correlations across three distinct loci in chromosomes 1 (**a**), 4 (**b**) and 7 (**c**). ** = significant correlations ($p$-value < 2.629e−05); · = nominally significant correlations; ns = non-significant correlations ($p$-value ≥ 0.05). Absence of correlations means that the bivariate test was not performed (See Methods for details). CD Crohn's disease, MS Multiple sclerosis, UC Ulcerative colitis, PD Parkinson's disease. Source data found in Supplementary Data 1.

is the nearest gene at a disease risk locus for PD[7], whereas *NEK1* within this locus is a known ALS risk gene[5]. Finally, there was a significant correlation at a locus [chr7:49632427-50894508] that includes the gene *IKZF1*, which encodes a transcription factor of the zinc-finger DNA-binding protein family, involved in B cell activation and differentiation, between AD and MS ($r_g = -0.77$; $p$-value = 3.55e−06). This same locus was also nominally positively correlated between AD and CD ($r_g = 0.489$; $p$-value = 0.002) (Fig. 2; Supplementary Fig. 4).

The Human Leukocyte Antigen (HLA) locus, a region in the genome with clear immune influences, spans multiple genomic loci that were tested in the analysis. However, we only observed one locus within this region [chr6:32208902-32454577] with significant positive correlations between neurodegenerative and immune-mediated diseases, correlated between AD and MS ($r_g = 0.778$; $p$-value = 2.01e−06). This locus contains the Major Histocompatibility Complex (MHC) class II gene *HLA-DRA*. The same genomic locus was positively correlated between SCZ and UC ($r_g = 0.683$; $p$-value = 1.47e−06). An additional locus

also spanning the HLA region [chr6:32682214-32897998] was positively correlated between SCZ and UC ($r_g = 0.626$; $p$-value = 9.99e−06) and between CD and SCZ ($r_g = 0.677$; $p$-value = 2.58e−05). This locus includes the following MHC class II genes: *HLA-DQA2*, *HLA-DQB2* and *HLA-DOB*.

**Gene expression levels of immune-related genes share causal signals with neurodegenerative diseases.** We then moved forward to ask whether regional genetic correlations could allow us to identify immune targets for neurodegenerative diseases. To do so, we estimated regional genetic correlations between diseases and genes significantly expressed across seven immune cell types (i.e., naive B cells, memory B cells, classical monocytes, CD4+ naive T cells, CD8+ naive T cells, CD4+ effector memory T cells and CD8+ effector memory T cells) from one of the largest datasets, the OneK1K dataset[22], to assess if changes in gene expression are correlated with disease risk. We performed 1628 pair-wise correlations across 2553 significantly expressed genes in

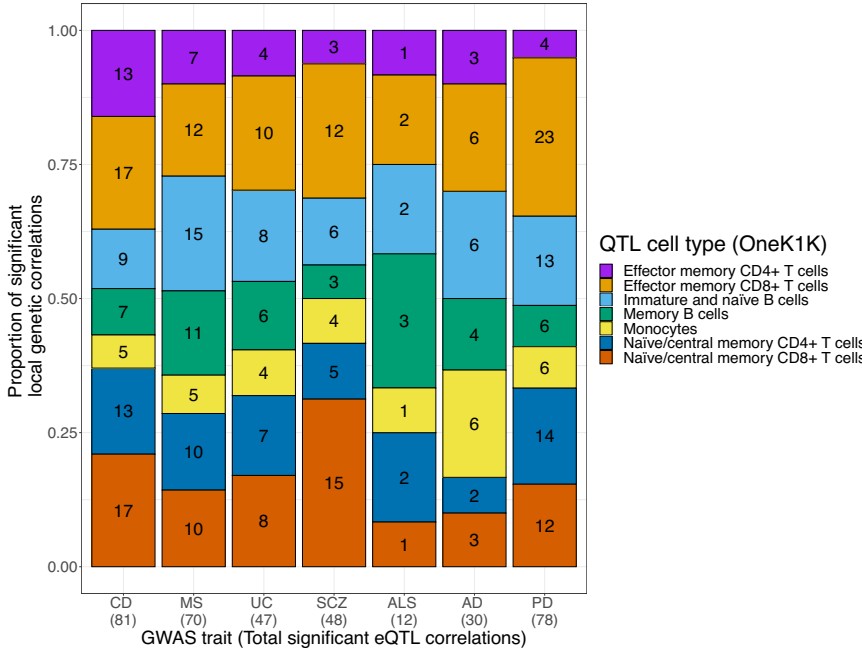

**Fig. 3 Proportion of significant regional genetic correlations between diseases and gene expression, across seven immune cell types.** The numbers in the bar plots indicate the absolute number of significant correlations observed. There were no significant differences in the proportion of significant correlations across traits for each of the cell types tested (Chi-square test: $p > 0.05$). As there were no significant correlations for neither frontotemporal dementia nor Lewy body dementia, those traits are not displayed. CD Crohn's disease, MS Multiple sclerosis, UC Ulcerative colitis, SCZ Schizophrenia, ALS Amyotrophic lateral sclerosis, AD Alzheimer's disease, PD Parkinson's disease. Source data found in Supplementary Data 2.

at least one cell type (Supplementary Table 2), which resulted in 366 significant correlations (FDR < 0.01) (Fig. 3; Supplementary Data 2). In addition, we followed up on the significant correlations through colocalization analyses to assess if there is a shared causal signal driving the correlation. This information provided insights about specific immune cell types and genes implicated in disease risk.

Across the tested neurodegenerative diseases, there were no expressed genes significantly correlated with FTD or LBD, which were the two GWASes with the smallest sample size. In terms of the total number of tested correlations within a disease, qualitatively, AD had a higher proportion of correlations with expressed genes in classical monocytes. Similarly, qualitatively there was a relatively higher proportion of expressed genes significantly correlated with ALS in memory B cells (but there were only 12 significant correlations across all cell types for ALS). Finally, compared to all other tested diseases, there was a qualitatively relatively higher proportion of expressed genes significantly correlated with PD in CD8+ effector memory T cells. We tested whether the proportion of significant correlations per a particular trait-cell type combination was significantly different across the total number of significant correlations per trait, using the prop.test() function in R. Quantitatively, we did not reject the null hypothesis for any cell type proportion (chi-square test $p > 0.05$), and thus cannot conclude that the proportions are significantly different (Fig. 3). Of note, the absolute number of significant correlations was not solely driven by the number of significant GWAS signals. For instance, SCZ is highly polygenic, but CD was the disease with the highest number of significant correlations. These results provide an initial overview of how disease risk across neurodegenerative diseases may be influenced by different immune cell types.

Significant correlations between diseases and expressed genes were distributed across all autosomes except on chromosome 9, where we only observed nominally significant correlations

(Supplementary Fig. 5). In addition, there were genes for which their expression was significantly correlated with a disease across more than one cell type ($N = 47$), whereas other expressed genes were significantly correlated with a disease in only one cell type ($N = 96$). For example, the expression of *BIN1* was positively correlated with AD across five immune cell types (i.e., memory B cells, CD4+ naive and effector memory T cells and CD8+ naive and effector memory T cells). It was also nominally correlated with LBD only in CD4+ effector memory T cells (Fig. 4a). *BIN1* is ubiquitously expressed across multiple tissues, including the brain, and is implicated in AD pathogenesis, possibly through its role in neuron hyperexcitability[23]. However, *BIN1* expression in B cells has not been associated with AD risk. Our colocalization analysis in the *BIN1* region indicate that there is no colocalization between AD risk and gene expression (H3 > 0.99). This result suggests that different variants in the locus influence either *BIN1* expression in B cells or AD risk (Fig. 5), highlighting the importance of complementing significant regional genetic correlations with colocalization analyses.

Amongst all diseases assessed, we observed a relatively higher number of significant correlations between PD and gene expression across all cell types tested ($N = 79$), many of which were not correlated with other diseases (Supplementary Fig. 5). After following up these significant correlations, we observed colocalization (H4 > 0.8) with three genes expressed in one or more cell types (i.e., *RAB7L1, ARSA* and *KANSL1-AS1*) (Fig. 5; Supplementary Data 3).

The expression of *RAB7L1* in CD4+ naive T cells was positively correlated with PD ($r_g = 0.826$; $p$-value = 0.001). This gene is a known risk locus for PD[6,7], involved in the regulation of the T cell receptor signalling pathway. It has also been shown to interact with *LRRK2* to alter the intraneuronal sorting of proteins and the lysosomal pathway[24,25], suggesting that the overexpression of *RAB7L1* in T cells may increase PD risk through the interaction with *LRRK2*.

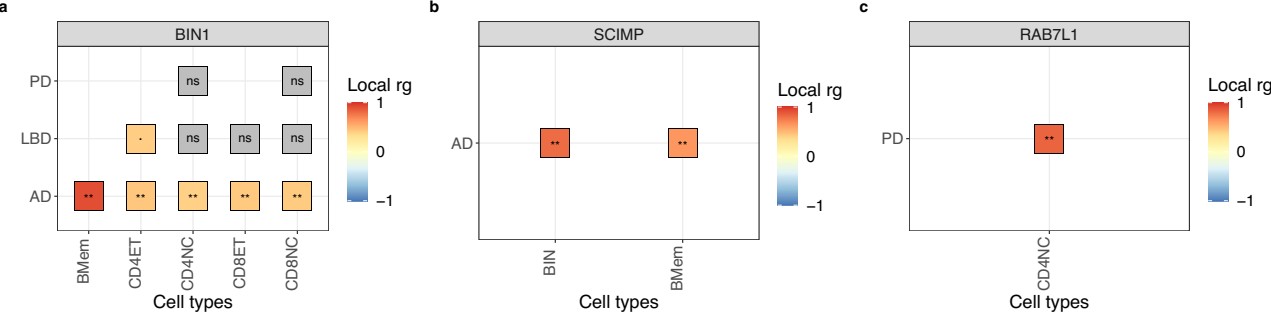

**Fig. 4 Examples of significant regional correlations between gene expression levels and diseases across three genes. a** *BIN1* (chromosome 2) was significantly correlated with AD across various adaptive immune cell types. **b** *SCIMP* (chromosome 17) was significantly correlated with AD across B cells. **c** *RAB7L1* (chromosome 1) was significantly correlated with PD in naive CD4 +T cells. ** = significant correlations (FDR < 0.01); · = nominally significant correlations; ns = non-significant correlations (*p*-value ≥ 0.05). AD Alzheimer's disease, LBD Lewy body dementia, PD Parkinson's disease, BMem Memory B cells, CD4ET CD4+ effector memory T cells; CD4NC CD4+ naive T cells; CD8ET CD8+ effector memory T cells; CD8NC CD8+ naive T cells. Source data found in Supplementary Data 2.

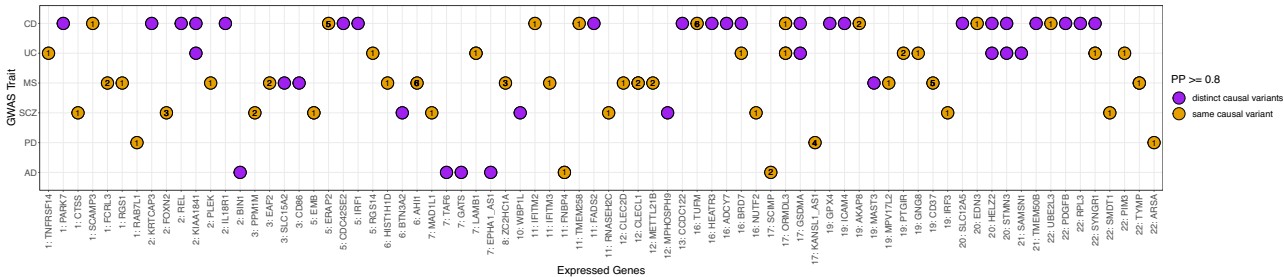

**Fig. 5 Colocalization results between diseases and expressed genes in at least one immune cell type.** Pairs of traits that share a causal variant through colocalization analysis are highlighted in orange (Posterior Probability, PP H4 ≥ 0.8), whereas pairs of traits that have distinct causal variants are highlighted in purple (PP H3 ≥ 0.8). The number inside the colocalized signals indicates the number of cell types for which a colocalization was observed. The cell type(s) for each disease-gene pair displayed here are listed in Supplementary Data 3.

The expression of *KANSL1-AS1*, an anti-sense RNA gene, was negatively correlated with PD across all adaptive immune cell types, but the correlation was the strongest for CD8+ T cells (i.e., effector memory T cells: $r_g = -0.831$; *p*-value = 1.34e−39, naive T cells: $r_g = -0.768$; *p*-value = 1.13e−25; Supplementary Fig. 5). In addition, colocalization analysis suggested the presence of a shared causal variant at the *KANSL1-AS1* locus (Fig. 5; Supplementary Data 3). The protein coding gene *KANSL1* is in the *MAPT* locus, which has been previously associated with PD[6,7,26,27], but recent experimental evidence suggests that the differential expression of another gene in the *MAPT* locus, *KANSL1*, also plays a crucial role in PD risk[28].

Of the initial 366 significant correlations observed across all tested diseases traits and cell types, 92 correlations (25.14%) implicated loci that did not encompass genome-wide significant GWAS variants (*p*-value ≥ 5e−08). However, 33.7% of these aforementioned loci are suggestive of association (*p*-value < 1e−06), whereas the remaining 66.3% loci are nominally significant (*p*-value < 0.05) (Supplementary Data 4). We observed colocalization with only two of these loci: (1) between AD and the expression of *FNBP4* in memory B cells (H4 = 0.913) and in CD8+ T cells (H4 = 0.85 and 0.84, for effector and naive CD8+ T cells, respectively), and (2) between PD and the expression of *ARSA* in CD8+ effector T cells (H4 = 0.88). *FNBP4* (situated ~15,000 base pairs away from *CELF1*) has been previously identified as an AD risk locus[29], but in a more recent transcriptome-wide association study (TWAS) of AD, this gene was discarded in conditional analyses[30]. *ARSA* has been previously investigated as a PD risk locus in a Chinese population, in which no significant associations were found with

PD susceptibility[31]. These results provide an complementary line of in silico evidence, suggesting that the expression of *FNBP4* and *ARSA* in adaptive immune cell types may play a role in AD and PD risk, respectively.

**Regional correlations with blood protein levels provide evidence of additional mechanisms involved in disease risk.** Proteins contain biologically meaningful information that cannot always be identified by solely assessing the transcriptome. For instance, as the proteome is often dysregulated by diseases, it is amenable to drug targeting and thus a better understanding of the of the proteome could aid in identifying novel treatments[32]. Therefore, we performed regional genetic correlations between diseases and protein levels in plasma using a large pQTL database[32], with the aim of exploring an additional level of biological variation and its relation to neurodegenerative disease risk.

We performed a total of 1863 bivariate tests between diseases and protein levels. We considered a significant regional correlation if FDR < 0.01. We observed significant correlations between protein levels and all diseases, except FTD. PD had a higher number of significant correlations, compared to other tested diseases (Supplementary Fig. 6; Supplementary Data 5). We evaluated the concordance between the regional genetic correlations performed with gene expression levels from diverse immune cell types and regional genetic correlations performed with protein levels derived from peripheral blood samples (Supplementary Fig. 7). A total of 68 unique gene/proteins were evaluated across both datasets (i.e., 68 genes with significant eQTLs also had genome-wide significant pQTLs), resulting in 35 genetic correlations that were at least nominally significant across

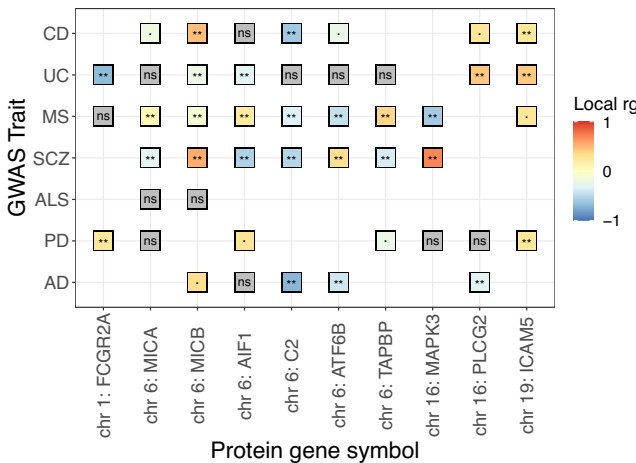

**Fig. 6 Regional correlations between protein levels and diseases shared between at least one test disease (i.e., AD, PD, ALS, LBD, FTD, SCZ) and at least one control disease (i.e., MS, UC, CD).** ** = significant correlations (FDR < 0.01); · = nominally significant correlations; ns = non-significant correlations ($p$-value ≥ 0.05). Source data found in Supplementary Data 5.

both tests, including 24 significant correlations across both tests (FDR < 0.01). The direction of effect was consistent across 17 of the significant correlations.

To obtain a biological understanding of the significant correlations with protein levels, specifically to assess if immune-related pathways were significantly enriched, we performed a gene-set enrichment test with FUMA[33]. We observed enrichment of several gene ontology (GO) biological processes (BP) across immune-mediated diseases, as well seven GO BP enriched for LBD (Supplementary Data 6). The enriched GO BP for UC, CD and MS include several immunological processes (such as adaptive immune response, regulation of immune effector process and regulation of immune system process), whereas the GO BP enriched for LBD correspond to gene-sets related to triglyceride processes (including protein lipid complex assembly and lipid complex subunit organization). While there was a partial overlap of the GO BP among the three immune-related diseases (5.5%), there was no overlap of GO BP between LBD and other diseases (Supplementary Fig. 8). Nevertheless, there were proteins harbouring significant correlations with at least one neurodegenerative disease and at least one of the immune-mediated diseases, none of which were significant in the regional genetic correlations with sc-eQTLs, but which have a function in the immune system (Fig. 6). However, we note that our regional genetic correlations are dependent on the data at hand and the current capacities of the computational approach. For instance, these results could change as GWAS sample sizes continue to increase, as additional sc-eQTL data become available, and once other types of variants (such as rare variants) that are not currently handled by LAVA are able to be assessed for regional genetic correlation using summary statistics.

One of the proteins harbouring a significant correlation with both a neurodegenerative and an immune-mediated disease was Fc fragment of IgG receptor IIa (*FCGR2A*), a cell surface receptor found on phagocytic cells (i.e., neutrophils, macrophages), involved in the process of clearing immune complexes[34]. Protein levels of *FCGR2A* were negatively correlated with UC ($r_g = -0.675$; $p$-value = 1.44e−20) and positively correlated with PD ($r_g = 0.237$; $p$-value = 6.02e−13) (Fig. 6). In line with these results, our regional genetic correlations among diseases highlighted a locus on chromosome 1 that encompasses this gene as

negatively correlated between PD and UC (Fig. 2a). However, there were no significant correlations between the expression of *FCGR2A* and PD or UC, even though the gene harbours genome-wide significant eQTLs in CD8+ effector memory T cells, but not in monocytes (macrophage precursors). These findings suggest that the protein levels of *FCGR2A* have opposite risk effects in UC and PD, which may be regulated by transcriptome-independent processes. Alternatively, *FCGR2A* may be differentially expressed in another cell type (such as macrophages), which we did not assess in the current study.

The protein levels of Phospholipase C gamma 2 (*PLCG2*) were significantly correlated with UC and AD, in opposite directions ($r_gUC = 0.442$; $p$-value = 2.28e−06, and $r_gAD = -0.491$; $p$-value = 3.03e−04) (Fig. 6). We did not estimate regional genetic correlations between the expression of *PLCG2* and diseases, given that this gene did not harbour genome-wide significant eQTLs in the immune cell types tested. Nonetheless, mutations in the gene *PLCG2* have been associated with dysregulation of the immune system, as well as with several dementias, in which distinct genetic variants are associated with different diseases, based on the identification of different functional point mutations across diseases[35]. For instance, the G allele of a missense variant within *PLCG2* has been shown to be protective against AD, LBD and FTD[36]. In contrast to our in silico observations of low *PLCG2* blood protein levels correlated with high AD risk, a recent study showed an upregulation of *PLCG2* expression in post-mortem brains of late-onset AD patients and its association to inflammation in microglia[37]. These seemingly discordant findings may be explained by different effects of *PLCG2* across stages of neurodegeneration, as well as by differences across sampled tissues.

## Discussion

The aim of the work was to assess the role that peripheral immune cells and related processes play in neurodegenerative diseases. We addressed this aim through orthogonal bioinformatics approaches: (i) by applying regional genetic correlations to relate neurodegenerative diseases to diseases known to be driven by immune dysfunction, (ii) by extending the correlation analysis by incorporating single-cell eQTLs to identify known gene-disease relationships in immune cell types, and (iii) by assessing the evidence for specific genes through expression and pQTL analyses. Through our approach, we identified links that warrant additional follow-up to better understand immune-mediated loci that may play a role in neurodegenerative diseases, such as the role of *SCIMP* expression in memory B cells as an AD risk locus, and the role of *FCGR2A* blood protein levels, correlated with PD risk.

By performing regional genetic correlations between pairs of diseases, we highlighted relationships between neurodegenerative diseases across loci encompassing known risk genes (e.g., *BIN1*, *TMEM175*, *APOE*). In addition, we highlighted relationships between neurodegenerative and immune-mediated diseases, suggesting the presence of shared immune-related biological pathways across these diseases (e.g., *FCGR2A*, *CLCN3*, *IKZF1*). The gene *FCGR2A*, for instance, located within a locus significantly correlated between PD and UC, has been previously associated with immune-mediated diseases[21], and is significantly expressed in CD8+ effector T cells and memory B cells[22]. Although our regional genetic correlations with gene expression levels indicated that the expression of *FCGR2A* is correlated with neither PD nor UC risk, we observed significant correlations with the corresponding protein levels for both PD and UC. Similarly, the gene *IKZF1* is within an LD locus significantly correlated between MS and AD, and is significantly expressed in CD8+

naive T cells[22]. However, regional genetic correlations with gene expression levels showed no evidence of significant correlation with AD ($r_g = 0.113$; $p$-value = 0.487), whereas the correlation test with MS was not performed due to lack of significant univariate signal. In addition, we observed significant positive correlations between AD and *SCIMP* expression in naive and memory B cells (Fig. 4b). Furthermore, colocalization analysis supported the hypothesis of a single shared causal variant (H4 = 0.99 and 0.85 for naive and memory B cells, respectively), suggesting that the expression of *SCIMP* in B cells may contribute to AD risk. *SCIMP* is a gene that has been previously associated with immune-mediated diseases, such as lupus and rheumatoid arthritis[38], as well as with AD risk[1,4]. The gene encodes a protein expressed in antigen-presenting cells, localized in the immunologic synapse, and serves as a regulator of antigen presentation[39]. Overall, this result supports a role of the adaptive immune system in AD risk, specifically of B cells, mediated by the expression of genes, such as *SCIMP*.

The observations at these loci (i.e., *FCGR2A* and *IKZF1*) suggest that different cell types or alternative molecular mechanisms may be involved in disease risk. In fact, we observed only a partial overlap and concordance of loci evaluated in both gene expression and protein regional genetic correlations, in line with our expectations, given that the eQTL and pQTL datasets used were generated through different sources: single-cell RNA-sequencing from specific immune cell types and bulk blood tissue, respectively.

Our regional analysis of genetic correlations with gene expression levels shed light on overall differences among diseases, including the varying proportion of correlations accounted for by gene expression in immune cell types. For example, a higher proportion of significant correlations between AD and sc-eQTLs were accounted for by genes expressed in classical monocytes, including genes within and outside of the HLA region. These results are in line with previous evidence pointing at a key role of the innate immune system (i.e., microglia) in AD. However, it has also been suggested that circulating monocytes participate in the clearance of Aβ plaques that diffuse into the bloodstream[40,41], and that monocyte-derived macrophages have a more efficacious phagocytic capacity than microglia in the brain[42,43]. Therefore, aside from the crucial role microglia play in AD, peripheral innate immune cells may be independently contributing to AD risk via changes in transcription levels. The communication between both the central and peripheral components are likely required for healthy brain function, as previously reviewed, and a breakdown in this cross-talk may contribute to neurodegenerative disease risk[44]. This interaction between both immune system components is also supported by a recent study in mouse models for AD where T cells, in cross-talk with microglia, promote neurodegeneration[45].

Our gene set enrichment analysis, aimed at better understanding the significant correlations observed between diseases and protein levels, did not highlight significant immune-related biological pathways enriched for neurodegenerative diseases. However, we identified proteins across neurodegenerative and immune-related diseases for follow-up. We believe that novel pathway enrichment methods that consider gene-specific weights (i.e. weights dependent on a measure of the strength of the regional correlations) could provide an important avenue for follow-up, alongside the current methods that consider all genes as input to the analysis as having equal effects.

Through our data-driven approach, we provide fine resolution links for genomic regions to a disease in a particular cell type to better understand the etiology of neurodegenerative diseases in relation to the peripheral immune system. However, this approach is not without limitations. First, our analyses used

GWAS and QTL datasets of inferred European genetic ancestry, which is a limitation stemming from the lack of diversity in GWAS cohorts[46]. Genetic ancestry may be particularly important for immune function given different selection pressures placed by infectious diseases. Even though GWAS of neurodegenerative diseases have been performed in cohorts of other genetic ancestries[5], the sample sizes needed to reach sufficient power to identify significant correlations falls short, as we observed in the case of the smaller FTD and LBD GWAS datasets. Second, it is known that there are sex differences in the incidence of some neurodegenerative and immune-mediated diseases, but the GWAS datasets used do not include sex-stratified analyses or sex chromosome data, which is a limiting factor in the identification of (i) sex-specific or sex-skewed expressed genes, or (ii) candidate immune-related genes on the sex chromosomes. Future studies that consider sex-chromosomes or sex differences may provide new insights on underlying mechanisms or cell types involved in disease pathogenesis. Third, mechanisms other than varying gene expression could be responsible for the absence of particular eQTL-disease correlations, which we were not able to capture with our approach. One such example is the absence of genome-wide significant eQTLs for *LRRK2* in any of the immune cell types tested, a gene in which missense point mutations have been associated with PD risk[47]. Indeed, there have been reports suggesting that *LRRK2* does not only influence at the transcriptional-level, but also at the protein-level rather. For instance, it has been reported that *LRRK2* protein levels are increased in individuals with sporadic PD and other studies have observed that *LRRK2* transgenic mice exhibit dysregulated immune responses[48–50]. An additional consideration that was not assessed here, but will be important to further this research, is to incorporate data from longitudinal clinical studies to better evaluate peripheral and central immune mechanisms over time[44]. Finally, our main analyses are based on correlations, which cannot assess causal relationship between diseases and the molecular mechanisms assessed. Nonetheless, we have highlighted immune-related genes as clear candidates for further investigation to better understand neurodegenerative diseases.

Here, we chose to focus on the role of the immune system in neurodegeneration. Indeed, for certain neurodegenerative diseases, the role of the immune system is arguably more prominent than for others. For AD, for instance, there is substantial evidence of the role of the immune system, including immune-mediated tissues being enrichment for heritability and single-cell RNA-sequencing enrichment analyses pointing to microglia[4,51]. Work has also highlighted activated microglia and T cell responses in tauopathies[45]. Studies of other neurodegenerative diseases have identified the HLA region on chromosome 6 as being implicated in disease risk; for example, the HLA region is a genome-wide signal in PD GWASes[6,7], and there is evidence of immune-mediated genetic enrichment for FTD within this region[11]. Furthermore, for PD, there is recent converging evidence of immune-related influences on disease risk through the role of T cells in brain inflammation and neurodegeneration[52,53]. Altogether, these lines of evidence, in combination with the utility of repurposing existing immune-based treatments, motivated our focus on immune-mediated factors presented here. Nevertheless, we acknowledge evidence of enrichment in non-immune cell and tissue types across neurodegenerative diseases. For instance, for PD and ALS, analyses have demonstrated heritability enrichment for brain tissue, and work based on single-cell RNA-sequencing has shown enrichment for neurons for these two diseases[5,7]. Likely, neurodegenerative disease risk and progression are mediated by a complex interplay of multiple cell and tissue types influencing a diverse set of biological systems. Neurodegeneration is not limited to a single system such as the central nervous

system or, as assessed here, the immune system. Indeed, future research is warranted to further investigate immune-mediated mechanisms on neurodegenerative disease risk, and to expand explorations to assess other biological systems to improve our understanding of additional disease mechanisms.

## Methods

**Datasets and data formatting**. We obtained genome-wide association study (GWAS) datasets from publicly available repositories, or requested access to their corresponding summary statistics. We selected five GWAS datasets from European genetic ancestry case/control studies of common neurodegenerative diseases as test traits: (1) Alzheimer's disease, (2) Parkinson's disease[7], (3) Lewy body dementia[19], (4) amyotrophic lateral sclerosis[5], and (5) frontotemporal dementia[3]. We also included three GWAS datasets corresponding to case/control studies of immune-mediated diseases as control traits: (1) multiple sclerosis[54], (2) ulcerative colitis[55], and (3) Crohn's disease[55]. Finally, we included as a test dataset a well-powered case/control study of schizophrenia[56], a neuropsychiatric disorder in which there is a genome-wide association with the Human Leukocyte Antigen (HLA) region, encoding genes that play a key role in the immune system. Detailed information on the GWAS sample sizes, number of genetic variants, genomic build, and source URLs are available in Table 1. After download, we formatted the GWAS summary statistics with R (version 4.0.2)[57] and lifted over the genomic coordinates to the Human Genome Build GRCh37 with the R package *rutils* version 0.99.2[58] as needed. We used the R package *SNPlocs.Hsapiens.dbSNP144.GRCh37*[59] to map reference SNP IDs (rsids) to genomic coordinates or vice versa. All analyses in the present study were performed using the Digital Research Alliance of Canada compute clusters.

We obtained single-cell expression quantitative trait loci (sc-eQTLs) summary statistics from the OneK1K study[22] by personal communication with the corresponding author. The dataset includes single-cell expression data on 1.27 million peripheral blood mononuclear cells in 982 individuals of European genetic ancestry, clustered into 14 immune cell types. To minimize the multiple testing burden, we selected a subset of these cell types for the present study. Specifically, we included the following cells from the innate and adaptive immune system: (1) classical monocytes, (2) effector memory CD4+ T cells, (3) naive CD4+ T cells, (4) effector memory CD8+ T cells, (5) naive CD8+ T cells, (6) naive B cells, and (7) memory B cells. To explore an additional level of biological variation, we also obtained summary statistics of plasma protein QTLs, pQTLs[32], corresponding to the "European American" sample, including 7213 individuals (http://nilanjanchatterjeelab.org/pwas/).

All ethical regulations were followed and informed consent was obtained in the original manuscripts. The relevant local institutional review boards approved the studies.

**Statistics and reproducibility**. Code to reproduce analyses have been provided (See Data Availability statement). Sample sizes of the datasets are described in Table 1. There were no replicates.

**Genome-wide genetic correlations across GWAS datasets**. We estimated genome-wide genetic correlations ($r_g$) across GWAS trait pairs using linkage disequilibrium score regression (LDSC)[16]. We first formatted GWAS summary statistics for each trait using the *munge_sumstats.py* function to align the alleles and keep SNPs present in the HapMap Project Phase 3, with the MHC region removed. Next, we ran the *ldsc.py* function for each trait pair using the 1000 Genomes Project Phase 3 European super-population as the LD reference to obtain $r_g$ estimates. We applied a Bonferroni-corrected p-value threshold to account for the number of pair-wise correlations performed, and subsequently defined a significant correlation if p-value < 0.0014.

**Regional genetic correlations across GWAS datasets**. We estimated regional genetic correlations ($r_g$) across GWAS pair traits with the R package *Local Analysis of [co]Variant Association* (LAVA)[18]. In brief, LAVA conducts a bivariate test to assess pairwise $r_g$ across predefined genomic regions. It is not limited to two GWAS traits, but other genome-wide associations, such as quantitative trait loci for gene expression or protein levels (as we describe in the subsequent section), can be used. A significant bivariate test suggests that there is a statistically significant genetic correlation at the tested region for the pair of traits. For each trait, LAVA can also assess univariate regional genetic signal (i.e. an estimate of the per-trait local heritability), which can then be used to filter out regions with sufficient univariate signal to be subsequently assessed in the bivariate test. We used the genomic regions defined as autosomal LD blocks ($N = 2495$) across autosomal chromosomes by Werme et al., which are characterized by having minimum LD across regions, a minimum of 2500 variants included on each LD block, and with an average LD block size of 1 million bases. To define which genomic regions to test across GWAS traits, we selected LD blocks that contained at least one genome-wide significant signal in at least one GWAS trait ($n = 389$). We considered sample overlap across GWAS datasets in the analysis by including the pair-wise genetic covariance estimated by LDSC and further standardizing it into a correlation matrix. To estimate regional $r_g$, we first performed a univariate test for each trait

per LD block and performed a bivariate test only for those trait pairs that had a significant univariate genetic signal (p-value < 1.28e−04, correcting for the 389 LD blocks tested). We applied a Bonferroni-corrected p-value threshold to account for the number of pair-wise regional correlations performed and defined a significant correlation if p-value < 2.63e−05 (0.05/1902) (Supplementary Fig. 9).

**Regional genetic correlations between GWAS and QTLs**. We estimated regional $r_g$ between GWAS and QTL datasets using LAVA[18]. In the case of the regional correlations between GWAS and gene expression levels, we tested protein and non-protein coding genes harbouring at least one genome-wide significant sc-eQTL per cell type separately. We extended the tested region 100 kb upstream and downstream of the start and end positions of the gene, which is where the majority of the *cis*-eQTLs are located[60]. We followed the same approach when defining the genomic regions to test between GWAS and protein levels, in which we included proteins that harboured at least one genome-wide significant pQTL, and extended the tested region +/− 100 kb from the start/end gene coordinates of the respective protein. In both cases (i.e. sc-eQTLs and pQTLs) we assumed that there was no sample overlap between the GWAS and QTL datasets, which we believe is a reasonable assumption. We estimated regional $r_g$ as described above, in which we first performed a univariate test for each trait, and then performed a bivariate test between GWAS-QTL only if both had a significant univariate genetic signal, correcting for the number of genes or proteins tested (Supplementary Fig. 2). We applied an FDR correction to the p-value threshold to account for the number of pair-wise regional correlations, separately for the analysis with sc-eQTLs and pQTLs, thus defining a significant GWAS-QTL correlation if FDR < 0.01. We chose an FDR correction, instead of a stricter Bonferroni correction, given that genic regions do not necessarily represent unique regions of linkage equilibrium (i.e. variants in one gene may be in linkage disequilibrium with variants in nearby genes as well).

**Colocalization to follow-up on regional correlations between GWAS and gene expression levels**. We performed colocalization analysis to follow-up on the significant (FDR < 0.01) regional correlations between GWAS and gene expression levels using the tool, *coloc*[61,62]. For those traits where the sample minor allele frequency (MAF) was available, we checked the correlation between the MAF of the 1000 Genomes European super-population and the sample MAF, which was 0.99 in all cases. Similar to the regional correlations approach, we tested the genic region +/− 100 kb from the start/end gene coordinates and assumed a maximum of one causal signal per colocalization. We tested a total of 366 GWAS-QTL pairs using the default SNP priors ($p_1 = p_2 = 1e−04$ and $p_{12} = 1e−05$). We considered a region to colocalize between gene expression levels and a GWAS trait, if the posterior probability (PP) of H4 ≥ 0.8, which suggests a high probability of a shared causal signal between both traits.

**Gene set enrichment analysis to follow-up on regional genetic correlations between GWAS and protein levels**. We performed a gene set enrichment analysis with the GENE2FUNC tool implemented in FUMA[33] to aid in the interpretation of the regional genetic correlations between GWAS traits and protein levels. We analysed one GWAS trait at a time and included only genes with protein levels that were significantly correlated with that GWAS trait (FDR < 0.01). We used Ensembl version 92 and included the list of 4657 genes for which protein levels were assessed (hence the genes used in the pQTL analysis) as the background set of genes[32]. FUMA performs a hypergeometric test for gene set enrichment using gene set databases obtained from MSigDB, WikiPathways, and the GWAS Catalog. Of the available datasets, we focused specifically on Gene Ontology biological processes. We set a minimum threshold of overlapping genes with gene sets of ≥2 and used the Benjamin-Hochberg FDR multiple testing correction method (alpha = 0.05) to define enriched gene sets.

**Validation of AD and PD signals using GWAS without proxy cases**. The AD and PD GWAS used in these analyses included proxy cases (i.e., individuals who do not have the disease of interest, but have a close relative who does). The inclusion of proxy cases has been suggested as a useful means to increase case sample size, particularly for late-onset disorders such as AD and PD[63]. However, there are also concerns raised on the impact of proxies on heritability and careful diagnosis, which has been examined in the context of AD[64]. As a sensitivity analysis to verify that our results involving AD and PD are not primarily driven by possible spurious effects of the inclusion of proxy cases, we reperformed the analyses for significant findings for the regional genetic correlations with gene expression levels and with protein levels using AD and PD without proxy cases[2,65]. Validation results are described in Supplementary Note 1 (Supplementary Figs, 10–12 and Supplementary Tables 3, 4). In brief, the directions of effect were consistent between the primary (AD and PD GWAS with proxy cases) and sensitivity analyses (AD and PD GWAS without proxy cases) when assessing the significant regional correlations (for either gene expression or protein levels) detected in the primary analysis that had sufficient univariate signal in the sensitivity analysis.

**Table 1 Overview of the GWAS summary statistics used in the present study.**

| GWAS (acronym) | N cases | N controls | N variants | Chrom-osomes | Genome build | LDSC observed SNP $h^2$ (Z-score) | LDSC intercept (SE) | Reference | URL |
|---|---|---|---|---|---|---|---|---|---|
| Alzheimer's disease (AD) | 75,671 (52,791 proxies) | 397,844 | 10,687,077 | 1-22 | GRCh37 | 1.06 % (1.860) | 1.0964 (0.0702) | Schwartzentruber et al.[4] | http://ftp.ebi.ac.uk/pub/databases/gwas/summary_statistics/GCST90012001-GCST90013000/GCST90012877/ |
| Amyotrophic lateral sclerosis (ALS) | 27,205 | 110,881 | 10,461,755 | 1-22 | GRCh37 | 3.82 % (8.489) | 1.0274 (0.0075) | Van Rheenen et al.[5] | https://www.projectmine.com/research/download-data/ |
| Crohn's disease (CD) | 12,194 | 28,072 | 9,550,617 | 1-22 | GRCh37 | 44.59 % (8.847) | 1.0962 (0.0148) | de Lange et al.[55] | http://ftp.ebi.ac.uk/pub/databases/gwas/summary_statistics/GCST004001-GCST005000/GCST004132/ |
| Frontotemporal dementia (FTD) | 2,154 | 4,308 | 6,026,384 | 1-22 | GRCh37 | 8.08 % (1.135) | 0.9995 (0.0073) | Ferrari et al.[3] | https://ifgcsite.wordpress.com/data-access/ |
| Lewy body dementia (LBD) | 2,591 | 4,027 | 7,827,747 | 1-22 | GRCh38 | 14.81 % (1.806) | 1.0055 (0.0084) | Chia et al.[19] | http://ftp.ebi.ac.uk/pub/databases/gwas/summary_statistics/GCST90001001-GCST90002000/GCST90001390/ |
| Multiple sclerosis (MS) | 14,802 | 26,703 | 8,244,101 | 1-22, X, Y | GRCh37 | 29.33 % (10.863) | 1.0164 (0.0101) | IMSGC[54] | Requested access through https://imsgc.net/?page_id=31 |
| Parkinson's disease (PD) | 33,674 (18,618 proxies) | 449,056 | 12,076,399 | 1-22 | GRCh37 | 1.62 % (10.125) | 0.9806 (0.007) | Nalls et al.[7] | https://www.pdgenetics.org/resources |
| Schizophrenia (SCZ) | 40,675 | 64,643 | 8,064,799 | 1-22, X | GRCh37 | 42.17 % (28.113) | 1.0533 (0.0121) | Pardiñas et al.[56] | https://pgc.unc.edu/for-researchers/download-results/ |
| Ulcerative colitis (UC) | 12,366 | 33,609 | 9,567,780 | 1-22 | GRCh37 | 23.97 % (9.255) | 1.097 (0.0138) | de Lange et al.[55] | http://ftp.ebi.ac.uk/pub/databases/gwas/summary_statistics/GCST004001-GCST005000/GCST004133/ |

**Reporting summary**. Further information on research design is available in the Nature Portfolio Reporting Summary linked to this article.

## Data availability

The authors declare that all data supporting the findings of this study are available within the paper and its supplementary information files. Source data underlying Figs. 2–6 are presented in Supplementary Data 1,2,3 and 5. The GWAS summary statistics analysed during the current study are available through the NHGRI-EBI GWAS Catalog FTP site or consortium-specific websites. The source links for each file is provided in Table 1. OneK1K sc-eQTL summary statistics that include effect sizes and standard errors were provided through personal communication with the corresponding author of the paper describing this dataset. pQTL summary statistics are available from the Nilanjan Chatterjee lab webpage at http://nilanjanchatterjeelab.org/pwas/.

## Code availability

All code generated for performing the analyses in the present study is available in the following GitHub repository: https://github.com/GaglianoTaliun-Lab/neuroimmune_genetics_project and in the corresponding Zenodo repository: https://doi.org/10.5281/zenodo.8064546[66].

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

## Acknowledgements
We thank the participants from all cohorts who contributed to the study. This research was enabled in part by support provided by Calcul Québec (https://www.calculquebec.ca/) and the Digital Research Alliance of Canada (alliancecan.ca). SAGT was funded by a Junior 1 award from the Fonds de recherche du Québec - Santé (FRQS; https://frq.gouv.qc.ca) and by Operational Funds from the Institut de valorisation des données (IVADO; https://ivado.ca). SAGT and FLD acknowledge support from a Canadian Institutes of Health Research (CIHR) Project Grant (PJT 183817). S.W.S. was supported in part by the Intramural Research Program of the National Institute of Neurological Disorders and Stroke, National Institutes of Health (program #: ZIANS003154). M.R. was supported through the award of a UKRI Medical Research Council Clinician Scientist Fellowship (MRC Grant Code: MR/N008324/1).

## Author contributions
F.L.D., S.A.G.T., S.W.S. and M.R. conceived and designed the study. F.L.D. analysed data and drafted the figures. R.H.R. and F.L.D. implemented the computer code. S.A.G.T. supervised the research. S.W.S. and M.R. provided clinical insight to data interpretation. F.L.D. and S.A.G.T. wrote the initial manuscript. All authors contributed to the critical analysis and revision of the manuscript.

## Competing interests
S.W.S. serves on the Scientific Advisory Council of the Lewy Body Dementia Association and the Multiple System Atrophy Coalition. She receives research support from Cerevel Therapeutics. All other authors declare no competing interests.
