## [Peer Review File · Communications Biology]

Reviewers' comments:

Reviewer #1 (Remarks to the Author):

OVERALL: This excellent study by Lona-Durazo et al performed a systematic assessed the potential role of the immune system in 5 neurodegenerative diseases (NDs) by establishing regional genetic correlations between NDs and immune-cell-derived sc-eQTLs, as well as regional genetic correlations between NDs and protein levels. The authors report significant regional genetic correlations between sc-eQTLs and NDs across 151 unique genes with representation by both the innate and adaptive immune systems across NDs examined. Finally, co-localization analyses suggested immune-related candidate causal risk genes and significant regional correlations with protein levels across 156 unique proteins. These findings are high-impact because they advance our understanding of the immune component of NDs with the short-term objective of repurposing existing immunotherapies to delay or slow progression of NDs. Appropriate methodologies and databases were employed for the analyses and the statistical analysis is in good shape.

A few points to address to strengthen the manuscript:

1. Given that associations were found with HLA-DRA, please comment on the lack of association with the HLA-DRA SNP previously reported by Hamza et al 2010.
2. The authors should comment as to why CLCN3 is associated with risk for PD and UC and not the other NDs.
3. The authors should comment on the significance of the association of PD risk with CD8+ memory pathways given that the human immunophenotyping data implicates CD4+ synuclein-specific T cells, not CD8 cytotoxic cells.
4. The authors should discuss the implications of the statement below. Given that the field has been guided by GWAS and eQTLs, what are we do make of this lack of concordance?

Nevertheless, there were proteins harbouring significant correlations with at least one neurodegenerative disease and at least one of the immune-mediated diseases, none of which 336 were significant in the regional genetic correlations with sc-eQTLs.

5. Given that there are differences in etiology and function between central and peripheral myeloid cells, the statement that macrophage protein profiling might be informative is spot on and should be the focus of follow-up studies by the authors with groups that are currently doing such analyses in colonic biopsies of PD, IBD, and controls (Tansy lab presentation at MJFF summits).

6. In the Discussion, line 400 mentions the potential independent contribution of peripheral immune cells to AD. However, research shows that the central and peripheral compartments communicate to keep the brain healthy and that this conversation likely breaks down in and may even be a contributor to neurodegenerative disease. This idea and the challenges to the field are elegantly reviewed by Bettcher and colleagues in a review from last year in Nature Reviews Neurology and is worth a citation here to direct the reader for more in-depth discussion.

7. The points raised in the discussion about sex-specific differences are good. But what is not mentioned and should be is the need for longitudinal studies because it is expected that immune system dysfunction will be dynamic and progressive and getting a single snap-shot in time is just the beginning. This is one of the key takeaways from the Bettcher review and again is worth citing for more in-depth info for the reader.

8. The lack of association with eQTLs with LRRK2 is not entirely surprising and the authors should not that at the protein level, it has been reported that LRRK2 protein levels are increased in people with sporadic PD (Cook et al NPJ PD) and there are more recent reports of LRRK2 mutant mice and mutant carriers with dysregulated immune responses so it is not all about the transcriptional level.

Reviewer #2 (Remarks to the Author):

Summary of the study:

In this study, Lona-Durazo and colleagues aimed at assessing the potential role of the immune system in five neurodegenerative diseases using bioinformatic approaches.

The authors used GWAS data (summary statistics) to investigate genome-wide as well as regional

genetic correlations across traits. Using single-cell RNA data and protein data, the authors estimated genetic correlations between GWAS traits and sc-eQTLs and pQTLs. The authors find several significant correlations at different levels: across GWAS traits, between GWAS traits and sc-eQTLs, and between GWAS traits and protein levels. They finally used colocalization techniques as well as gene-set enrichment analysis to fine-map their associations. Overall, the authors show that significant regional correlations exist between different neurodegenerative diseases and sc-eQTLs within the immune system. The manuscript is very well written and demonstrates high technical/statistical rigor.

Major comments

1. The methods are, in general, very well written. However, for "Regional genetic correlations across GWAS datasets" (line 105, Methods) I had to go through LAVA manuscript to fully understand what the algorithm does. I think it would be very helpful if the authors provide more information about what the software does. For example, the authors mentioned both the univariate test and the bivariate test, but they don't really explain how to interpret these tests (what does a significant association in univariate/bivariate tests mean?). The same, in my opinion, goes for the paragraph "Regional genetic correlations between GWAS and QTLs" (line 118, Methods). I think the authors should spend a few more sentences to guide the reader through these sections. This would improve readability and reproducibility. A graphical representation of the methods could also help.
2. In my opinion, the main outcome of the research needs to be sharpened: the authors find several potentially interesting results in their analyses, yet the abstract reports a summary overview of the findings that is not easy to interpret. Personally, I like the AD/PD-related results of the correlation of GWAS traits and sc-eQTLs. Should the authors put more attention to this in the abstract and discussion?
3. In the results section regarding sc-eQTLs proportions in the different traits (lines 250-256, Results), the authors repeatedly mention that the proportion of a given cell-type was higher or relatively higher in a trait. This information is well reported in Figure 3. However, I think the manuscript would gain from formally testing whether the proportions of a given cell-type are significantly higher compared to other cell-types and other traits. This could provide new information about which cell-types are more influential in which pathology.
4. In my opinion, the main figures could be improved. For example, in Figure 1, negative correlations are not reported in blue (as the legend shows), but rather in grey. I understand this is because the authors report in grey all non-significant correlations, however, I would stick to the colors in the legend (so red shades for positive correlations and blue shades for negative correlations). To indicate significance, authors could use stars, for example, ** for significant associations after multiple test corrections, and * for nominally significant associations. Figure 2 shows the regional correlation values for 3 regions in chromosomes 1, 4, and 7. Although the figure is informative, I think it could be improved. In my opinion, here the point is the presence of a shared regional GWAS signal across different traits, so I think it would be nice to visualize the actual SNP-associations in the region. This could be a (sort of) regional plot (e.g. fuma, locuszoom, snpxplorer) where GWAS is shown for the different traits (in different panels or different colors). This way it would be very clear that there is a shared signal.

Minor comments:

1. I think the section regarding sc-eQTLs in results, contains a lot of details that could be moved to the discussion section. For example, the information about what a specific gene does and what previous studies showed (for example in lines 274-278 for SCIMP, 293-295 for KANSL1, 302-308 about FNBP4 and ARSA): I think these sentences would fit better in the discussion section
2. The authors are very rigorous in correcting for multiple tests in all their analyses, which I appreciate a lot. In some analyses, they also use strict methods for corrections (e.g. Bonferroni), which may impact their results (as they also mention in line 204-211, Results). Did the authors try with less strict correction methods as well?
3. I like the fact that the authors share their code for reproducibility, however, I could not really access the GitHub page
4. In line 329-334, Results, the authors describe the results of the gene-set enrichment analysis. Although there are links to the supplementary materials, I think it would be helpful to show at least the most significant/representative GO terms here.
5. The authors mention that they conducted a sensitivity analysis for AD and PD regarding the use

of proxy-phenotypes in these GWAS (paragraph "Validation of AD and PD signals using GWAS without proxy cases", line 158, Methods). Results are properly reported in supplementary note. However, I think the authors should (briefly) report the outcome of the sensitivity analysis in the main text.

Reviewer #3 (Remarks to the Author):

Lona-Durazo and colleagues provide analyses on genome-wide association studies in neurodegenerative diseases (AD, PD, ALS, LBD, FTD), immune-mediated diseases (MS, CU, CD) and schizophrenia. First, they estimate (local) genetic correlations and subsequently between disease traits and eQTL in circulating immune cells and pQTL. They identify numerous correlations between disease traits and eQTL, of which a subset is probably explained by a shared causal variant (colocalization).

General comments:

The role of immune system in neurodegenerative diseases is of great interest. Dissection their relationship could inform therapeutic strategies that are highly needed in these progressive lethal diseases.

A strong point in this study is the use of latest available datasets (disease traits and eQTL) as well as including two positive controls (immune-mediated diseases and schizophrenia).

Two aspects of this study limited my enthusiasm. First, it is not clear how specific their results are to immune-cell-specific eQTL. The title states "highlight relationships between neurodegenerative diseases and the immune system", but what exactly is highlighted? Yes, sc-RNAseq data from immune-cells were used to estimate correlations and colocalization, but the eQTL might not be specific to immune-cells. In fact, many eQTL are shared across cell types and tissues. So, the observed eQTL might exert its effect on disease risk in a totally different cell-type or tissue. The authors do not show enrichment of immune-cell-specific eQTL in contrast to any other tissue, nor show experimental validation that disease risk is indeed mediated through immunological processes. By limiting eQTL to immune-cell derived eQTL the authors *chose* to highlight the immune system, disease biology (data) itself does not highlight the immune system.

My second point is in line with this. The authors do not clearly describe a converging mechanism or biological pathway that link immune processes to neurodegeneration. Rather, the study ends up presenting many trait-gene correlations either through eQTL or pQTL and leaving it up to the reader to pick interesting genes to follow up. In polygenic traits however, there are often many loci involved in disease risk and even within these loci there are many genes. Therefore, only highlighting individual trait-gene pairs is always risky due to the sheer amount of possible gene-trait pairs. In this context, it is important to know exactly how many tests have been performed or could have been performed to estimate the risk of false positives. Limiting the search space by "only looking at regions with a significant univariate association" while tempting, still increases the risk of false positives. Because the authors perform many similar analyses it is easy to lose the bigger picture when only selected gene-trait pairs are reported.

See the discussion on this topic by Lander and Kruglyak "Are whole-genome thresholds overly stringent?" in Lander & Kruglyak, Genetic dissection of complex traits: guidelines for interpreting and reporting linkage results, Nature Genetics, 1995.

Specific comments:

The table of included studies could be presented in the main text. Given the sample size is important for all of the downstream analyses, it could help the reader to have these easily findable. Also, could the authors provide the LDSC intercept next to the $h^2(\text{obs})$ estimate?

It might help to add $N=...$ to figure 1 to show that LBD and FTD studies are relatively underpowered for these analyses.

To limit the multiple testing burden the authors limit the search space for regional correlations. For example, between disease traits only regions with at least 1 genome-wide significant signal and significant univariate regional genetic signal in both traits are included. This limits the number of tests and leads to more lenient Bonferroni correction.

A similar filtering prior to correlation testing is applied for eQTL and pQTL.

The author could consider visualizing this in a flow chart so the reader can better interpret the exact filtering and reduction in number of tests in relation to all possible tests.

The results section states: "we confirmed the expected significant positive correlations across these diseases, providing internal validation of the robustness of our approach". What do the authors mean by "robustness of our approach" as these are results from published data with widely used and validated tools using a standard approach. Indeed, most of these expected positive correlations have been published using the exact same data and methods.

The authors highlight some regional genetic correlations between neurodegenerative diseases and mention the CBR4, NEK1, CLCN3, C4orf27, SH3RF1 locus in ALS, CD and PD. This locus was not fine mapped in Parkinson's disease (CLCN3 was only the nearest gene) and NEK1 is a known ALS risk gene in this locus, not CLCN3.

Figure 3 displays the relative number eQTL correlations per cell type on y-axis. For a better comparison the authors could add a panel with the absolute number of eQTL correlations to show number is lower in AD, PD and ALS. For extra clarity the total number of genome-wide significant loci could be described as these are higher in SCZ compared to AD/PD/ALS.

In line 299, what do the authors mean by a locus with a nominally significant p-value? Is this a SNP within this locus with $P < 0.05$? In that case, why is this relevant because (almost) every region has a SNP with $P < 0.05$ given the SNP density and multiple testing burden.

References are not formatted uniformly. Journal name, edition, page etc are often missing. For example ref 18 (journal missing), 22 (name misspelled), 26 (journal misspelled), 44 (pages missing). GWAS on ALS (van Rheenan et al) is cited twice (2016 and 2021)

We thank all three reviewers and the editor for providing valuable feedback on our manuscript and our work presented within it. Below we outline (in bold) how we have addressed each point in the revised version of the manuscript. We believe that these comments have improved our manuscript, and we look forward to your feedback.

Response to reviewers' comments:

Reviewer #1 (Remarks to the Author):

OVERALL: This excellent study by Lona-Durazo et al performed a systematic assessed the potential role of the immune system in 5 neurodegenerative diseases (NDs) by establishing regional genetic correlations between NDs and immune-cell-derived sc-eQTLs, as well as regional genetic correlations between NDs and protein levels. The authors report significant regional genetic correlations between sc-eQTLs and NDs across 151 unique genes with representation by both the innate and adaptive immune systems across NDs examined. Finally, co-localization analyses suggested immune-related candidate causal risk genes and significant regional correlations with protein levels across 156 unique proteins. These findings are high-impact because they advance our understanding of the immune component of NDs with the short-term objective of repurposing existing immunotherapies to delay or slow progression of NDs. Appropriate methodologies and databases were employed for the analyses and the statistical analysis is in good shape.

Thank you for reviewing our work!

A few points to address to strengthen the manuscript:

1. Given that associations were found with HLA-DRA, please comment on the lack of association with the HLA-DRA SNP previously reported by Hamza et al 2010.

Thank you for this comment. Hamza et al. 2010 (Nature Genetics) conducted a genome-wide association study and identified HLA-DRA as being significantly associated with Parkinson's disease (the top variant in the locus was rs3129882, a non-coding polymorphism in the first intron). Our regional genetic correlation analysis does not discredit this association, but rather simply suggests that given our current datasets and the current capabilities of the computational method used to estimate the correlation, Parkinson's and one of the other GWAS traits analysed were not significantly genetically correlated at the locus containing HLA-DRA.

2. The authors should comment as to why *CLCN3* is associated with risk for PD and UC and not the other NDs.

We found that *CLCN3* was significantly correlated between PD and UC ($r_g = 0.525$; p-value = $1.02e-05$) and nominally significant between ALS and CD ($r_g = 0.369$; p-value = 0.044). We have added a clarifying point that *CLCN3* is the nearest gene at one of the known risk loci for PD: "The gene *CLCN3* is the nearest gene at a disease risk locus for PD". Unfortunately, it is not straightforward to identify the exact reason(s) as to why

we did not find evidence of correlation with UC and other neurodegenerative disorders at the locus containing *CLCN3*. However, as PD was one of our larger GWAS in terms of sample size (see Table 1), certainly power is a potential explanation, which we acknowledge in the Discussion.

3. The authors should comment on the significance of the association of PD risk with CD8+ memory pathways given that the human immunophenotyping data implicates CD4+ synuclein-specific T cells, not CD8 cytotoxic cells.

Thank you for this point. We note that compared to all other tested diseases, there was a relatively higher proportion of expressed genes significantly correlated with PD in CD8+ effector memory T cells. However, this observation does not exclude findings with CD4+ T cells. Indeed, we found that *RAB7L1* (chromosome 1) was significantly correlated with PD in naive CD4+ T cells (Figure 4). Our results are limited by the data available through one of the largest single-cell datasets for immune cells used here from the OneK1K consortium. As we did not have access to CD4+ synuclein-specific T cells for testing local correlations, we unfortunately cannot comment on the implication of these cells in PD risk at specific loci based on the current analyses.

4. The authors should discuss the implications of the statement below. Given that the field has been guided by GWAS and eQTLs, what are we do make of this lack of concordance?

Nevertheless, there were proteins harbouring significant correlations with at least one neurodegenerative disease and at least one of the immune-mediated diseases, none of which 336 were significant in the regional genetic correlations with sc-eQTLs.

We have now added insight on this lack of concordance in the last section of the results: "However, we note that our regional genetic correlations are dependent on the data at hand and the current capacities of the computational approach. For instance, these results could change as GWAS sample sizes continue to increase, as additional sc-eQTL data become available, and once other types of variants (such as rare variants) that are not currently handled by LAVA are able to be assessed for regional genetic correlation using summary statistics."

5. Given that there are differences in etiology and function between central and peripheral myeloid cells, the statement that macrophage protein profiling might be informative is spot on and should be the focus of follow-up studies by the authors with groups that are currently doing such analyses in colonic biopsies of PD, IBD, and controls (Tansy lab presentation at MJFF summits).

Completely agree. Thank you so much for this point.

6. In the Discussion, line 400 mentions the potential independent contribution of peripheral immune cells to AD. However, research shows that the central and peripheral compartments communicate to keep the brain healthy and that this conversation likely breaks down in and may even be a contributor to neurodegenerative disease. This idea and the challenges to the field are elegantly reviewed by Bettcher and colleagues in a

review from last year in Nature Reviews Neurology and is worth a citation here to direct the reader for more in-depth discussion.

Thank you! We have added this reference to the discussion. “The communication between both the central and peripheral components are likely required for healthy brain function, as previously reviewed, and a breakdown in this cross-talk may contribute to neurodegenerative disease risk (ref: Bettcher et al. 2021). This interaction between both immune system components is also supported by a recent study in mouse models for AD where T cells, in cross-talk with microglia, promote neurodegeneration (ref: Chen et al. 2023).”

7. The points raised in the discussion about sex-specific differences are good. But what is not mentioned and should be is the need for longitudinal studies because it is expected that immune system dysfunction will be dynamic and progressive and getting a single snap-shot in time is just the beginning. This is one of the key takeaways from the Bettcher review and again is worth citing for more in-depth info for the reader. **Point noted. To the discussion we have added: “An additional consideration that was not assessed here, but will be important to further this research, is to incorporate data from longitudinal clinical studies to better evaluate peripheral and central immune mechanisms over time (ref: Bettcher et al. 2021).”**

8. The lack of association with eQTLs with LRRK2 is not entirely surprising and the authors should note that at the protein level, it has been reported that LRRK2 protein levels are increased in people with sporadic PD (Cook et al NPJ PD) and there are more recent reports of LRRK2 mutant mice and mutant carriers with dysregulated immune responses so it is not all about the transcriptional level. **We have now added a sentence on the protein-level influences of LRRK2 on the immune system to the discussion to address the lack of association with eQTLs and LRRK2: “Indeed, there have been reports suggesting that LRRK2 does not only influence at the transcriptional-level, but also at the protein-level rather. For instance, it has been reported that LRRK2 protein levels are increased in individuals with sporadic PD and other studies have observed that LRRK2 transgenic mice exhibit dysregulated immune responses (refs: Cook et al. 2017, Takagawa et al. 2018, Chang et al. 2022).”**

Reviewer #2 (Remarks to the Author):

Summary of the study:

In this study, Lona-Durazo and colleagues aimed at assessing the potential role of the immune system in five neurodegenerative diseases using bioinformatic approaches. The authors used GWAS data (summary statistics) to investigate genome-wide as well as regional genetic correlations across traits. Using single-cell RNA data and protein data, the authors estimated genetic correlations between GWAS traits and sc-eQTLs and pQTLs.

The authors find several significant correlations at different levels: across GWAS traits,

between GWAS traits and sc-eQTLs, and between GWAS traits and protein levels. They finally used colocalization techniques as well as gene-set enrichment analysis to fine-map their associations.

Overall, the authors show that significant regional correlations exist between different neurodegenerative diseases and sc-eQTLs within the immune system.

The manuscript is very well written and demonstrates high technical/statistical rigor.

Thank you for your review!

Major comments

1. The methods are, in general, very well written. However, for "Regional genetic correlations across GWAS datasets" (line 105, Methods) I had to go through LAVA manuscript to fully understand what the algorithm does. I think it would be very helpful if the authors provide more information about what the software does. For example, the authors mentioned both the univariate test and the bivariate test, but they don't really explain how to interpret these tests (what does a significant association in univariate/bivariate tests mean?). The same, in my opinion, goes for the paragraph "Regional genetic correlations between GWAS and QTLs" (line 118, Methods). I think the authors should spend a few more sentences to guide the reader through these sections. This would improve readability and reproducibility. A graphical representation of the methods could also help.

We thank the reviewer for this comment to aid in readability and reproducibility. We would like to note that to aid in reproducibility, we have made our code used to conduct this work publicly available on GitHub (https://github.com/GaglianoTaliun-Lab/neuroimmune_genetics_project). We have now provided more information on LAVA into the methods: "In brief, LAVA conducts a bivariate test to assess pairwise r_g across predefined genomic regions. It is not limited to two GWAS traits, but other genome-wide associations, such as quantitative trait loci for gene expression or protein levels (as we describe in the subsequent section), can be used. A significant bivariate test suggests that there is a statistically significant genetic correlation at the tested region for the pair of traits. For each trait, LAVA can also assess univariate regional genetic signal (i.e. an estimate of the per-trait local heritability), which can then be used to filter out regions with sufficient univariate signal to be subsequently assessed in the bivariate test."

2. In my opinion, the main outcome of the research needs to be sharpened: the authors find several potentially interesting results in their analyses, yet the abstract reports a summary overview of the findings that is not easy to interpret. Personally, I like the AD/PD-related results of the correlation of GWAS traits and sc-eQTLs. Should the authors put more attention to this in the abstract and discussion?

Thank you! We have now rephrased the abstract (while maintaining the 200 word count limit) to focus more on the AD and PD-related results. For instance, we include main findings of sc-eQTL correlations with PD in the abstract. We have tried to balance this comment to focus more on AD/PD-related results and Reviewer's 3 contrasting comment of presenting the overall picture.

3. In the results section regarding sc-eQTLs proportions in the different traits (lines 250-256, Results), the authors repeatedly mention that the proportion of a given cell-type was higher or relatively higher in a trait. This information is well reported in Figure 3. However, I think the manuscript would gain from formally testing whether the proportions of a given cell-type are significantly higher compared to other cell-types and other traits. This could provide new information about which cell-types are more influential in which pathology.

To address this point, we have now tested whether the proportion of significant correlations per a particular trait-cell type combination is significantly different. Specifically, we used the `prop.test()` function in R to test the null that the proportions in the groups are the same. None of the proportions were significantly different. The results from this statistical test have been added to the legend of Figure 3. We have now clarified in the text by specifying our observations as qualitative and adding in the results from `prop.test()`: “In terms of the total number of tested correlations within a disease, qualitatively, AD had a higher proportion of correlations with expressed genes in classical monocytes. Similarly, qualitatively there was a relatively higher proportion of expressed genes significantly correlated with ALS in memory B cells (but there were only 12 significant correlations across all cell types for ALS). Finally, compared to all other tested diseases, there was a qualitatively relatively higher proportion of expressed genes significantly correlated with PD in CD8+ effector memory T cells. We tested whether the proportion of significant correlations per a particular trait-cell type combination was significantly different across the total number of significant correlations per trait, using the `prop.test()` function in R. Quantitatively, we did not reject the null hypothesis for any cell type proportion (chi-square test $p > 0.05$), and thus cannot conclude that the proportions are significantly different (Figure 3).”

4. In my opinion, the main figures could be improved. For example, in Figure 1, negative correlations are not reported in blue (as the legend shows), but rather in grey. I understand this is because the authors report in grey all non-significant correlations, however, I would stick to the colors in the legend (so red shades for positive correlations and blue shades for negative correlations). To indicate significance, authors could use stars, for example, ** for significant associations after multiple test corrections, and * for nominally significant associations. Figure 2 shows the regional correlation values for 3 regions in chromosomes 1, 4, and 7. Although the figure is informative, I think it could be improved. In my opinion, here the point is the presence of a shared regional GWAS signal across different traits, so I think it would be nice to visualize the actual SNP-associations in the region. This could be a (sort of) regional plot (e.g. fuma, locuszoom, snpxplorer) where GWAS is shown for the different traits (in different panels or different colors). This way it would be very clear that there is a shared signal.

Thank you for this comment. We have updated Figure 1 (the global genetic correlation heat-map) with the suggestions from the reviewer. We have also created regional plots to show shared signals for the chromosomes 1, 4 and 7 regions presented in

Figure 2; so not to make this main Figure too dense, these plots are displayed in Supplementary Figures 3-5.

Minor comments:

1. I think the section regarding sc-eQTLs in results, contains a lot of details that could be moved to the discussion section. For example, the information about what a specific gene does and what previous studies showed (for example in lines 274-278 for SCIMP, 293-295 for KANSL1, 302-308 about FNBP4 and ARSA): I think these sentences would fit better in the discussion section.

We agree with the reviewer. We have moved the paragraph on SCIMP to the discussion. However, we have kept the other examples in the results as they flow directly from the following sentence in the results: “After following up these significant correlations, we observed colocalization ($H4 > 0.8$) with three genes expressed in one or more cell types (i.e., RAB7L1, ARSA and KANSL1-AS1) (Figure 5; Supplementary Table 3).”

2. The authors are very rigorous in correcting for multiple tests in all their analyses, which I appreciate a lot. In some analyses, they also use strict methods for corrections (e.g. Bonferroni), which may impact their results (as they also mention in line 204-211, Results). Did the authors try with less strict correction methods as well?

Thank you for this comment. We did not try with a less strict correction method, but we would like to note that our Bonferroni correction used for our regional genetic correlation analysis between GWAS traits is a more lenient Bonferroni. That is to say, we limited the search space for regional correlations to limit the number of tests and to have a more lenient Bonferroni correction. For instance, between disease traits only regions with at least 1 genome-wide significant signal and significant univariate regional genetic signal in both traits are included. Additionally, to account for the number of tests for the GWAS-QTL (either single cell eQTL or protein QTL), we applied an FDR correction to the p-value threshold to account for the number of pair-wise regional correlations, separately for the analysis with sc-eQTLs and pQTLs ($FDR < 0.01$). We chose an FDR correction, instead of a stricter Bonferroni correction for the GWAS-QTL correlations, given that genic regions do not necessarily represent unique regions of linkage equilibrium (i.e. variants in one gene may be in linkage disequilibrium with variants in nearby genes as well).

3. I like the fact that the authors share their code for reproducibility, however, I could not really access the GitHub page

The GitHub link should now be working https://github.com/GaglianoTaliun-Lab/neuroimmune_genetics_project

4. In line 329-334, Results, the authors describe the results of the gene-set enrichment analysis. Although there are links to the supplementary materials, I think it would be helpful to show at least the most significant/representative GO terms here.

After the reference to the Supplementary File (Supplementary File 3), we have now listed examples of the most significant GO terms in the main text: “The enriched GO BP for UC, CD and MS include several immunological processes (such as adaptive immune response, regulation of immune effector process and regulation of immune system process), whereas the GO BP enriched for LBD correspond to gene-sets related to triglyceride processes (including protein lipid complex assembly and lipid complex subunit organization).”

5. The authors mention that they conducted a sensitivity analysis for AD and PD regarding the use of proxy-phenotypes in these GWAS (paragraph “Validation of AD and PD signals using GWAS without proxy cases”, line 158, Methods). Results are properly reported in supplementary note. However, I think the authors should (briefly) report the outcome of the sensitivity analysis in the main text.

Thank you for this point. We have now summarized the results of the sensitivity analysis in the main text: “In brief, the direction of effects were consistent between the primary (AD and PD GWAS with proxy cases) and sensitivity analyses (AD and PD GWAS without proxy cases) when assessing the significant regional correlations (for either gene expression or protein levels) detected in the primary analysis that had sufficient univariate signal in the sensitivity analysis.”

Reviewer #3 (Remarks to the Author):

Lona-Durazo and colleagues provide analyses on genome-wide association studies in neurodegenerative diseases (AD, PD, ALS, LBD, FTD), immune-mediated diseases (MS, CU, CD) and schizophrenia. First, they estimate (local) genetic correlations and subsequently between disease traits and eQTL in circulating immune cells and pQTL. They identify numerous correlations between disease traits and eQTL, of which a subset is probably explained by a shared causal variant (colocalization).

Thank you for your assessment of our work!

General comments:

The role of immune system in neurodegenerative diseases is of great interest. Dissection their relationship could inform therapeutic strategies that are highly needed in these progressive lethal diseases.

A strong point in this study is the use of latest available datasets (disease traits and eQTL) as well as including two positive controls (immune-mediated diseases and schizophrenia).

Thank you for noting these strengths of our work.

Two aspects of this study limited my enthusiasm. First, it is not clear how specific their

results are to immune-cell-specific eQTL. The title states “highlight relationships between neurodegenerative diseases and the immune system”, but what exactly is highlighted? Yes, sc-RNAseq data from immune-cells were used to estimate correlations and colocalization, but the eQTL might not be specific to immune-cells. In fact, many eQTL are shared across cell types and tissues. So, the observed eQTL might exert its effect on disease risk in a totally different cell-type or tissue. The authors do not show enrichment of immune-cell-specific eQTL in contrast to any other tissue, nor show experimental validation that disease risk is indeed mediated through immunological processes. By limiting eQTL to immune-cell derived eQTL the authors *chose* to highlight the immune system, disease biology (data) itself does not highlight the immune system.

Thank you for raising this point.

With regard to the specificity of eQTLs to specific immune cells, we note that eQTLs can be common across all cell types. However, the genes themselves that are being regulated are often specific to the cell type. For example, in the case of immune cells and blood cell types in general, the expression profiles can be very distinctive – as shown in the 2015 GTEx flagship paper where it is shown that based on expression alone, blood samples clustered away from all other tissues tested. Therefore, one could have immune-cell type specific eQTLs because of the manner in which they are expressed in contrast to the regulating genetic variants. However, this scenario is likely not the case for all the genes we have highlighted, but applies for many. For example, *FCGR2A* or *IKZF1* are genes that are very cell type specific (for specific immune cells) and thus their eQTLs must also be. For genes such as these it is difficult to argue that disease risk would be mediated through non-immune processes. Additionally, the OneK1K sc-eQTL dataset used in the current study (specific for immune cells) is one of the largest in terms of sample size. Therefore, the possibility of using a negative control of non-immune cell types (for instance, such as sc-eQTLs derived from brain cells from the recent study from Bryois et al. 2022 *Nature Neuroscience*, with a sample size of 192 samples) from a different dataset would not be comparable in terms of sample size and power. This lack of comparability will be reflected in our regional correlation findings, making us unable to differentiate between a negative result versus a lack of statistical power. To summarize, yes, we have chosen to focus on immune cell derived eQTLs and the immune system. However, this decision is based on the growing body of literature supporting the role of immunity in neurodegeneration (e.g. Supplementary Figure 1): a role that we aimed to explore further using regional genetic correlation and the latest available high resolution datasets.

My second point is in line with this. The authors do not clearly describe a converging mechanism or biological pathway that link immune processes to neurodegeneration. Rather, the study ends up presenting many trait-gene correlations either through eQTL or pQTL and leaving it up to the reader to pick interesting genes to follow up. In polygenic traits however, there are often many loci involved in disease risk and even

within these loci there are many genes. Therefore, only highlighting individual trait-gene pairs is always risky due to the sheer amount of possible gene-trait pairs. In this context, it is important to know exactly how many tests have been performed or could have been performed to estimate the risk of false positives. Limiting the search space by “only looking at regions with a significant univariate association” while tempting, still increases the risk of false positives. Because the authors perform many similar analyses it is easy to lose the bigger picture when only selected gene-trait pairs are reported. See the discussion on this topic by Lander and Kruglyak “Are whole-genome thresholds overly stringent?” in Lander & Kruglyak, Genetic dissection of complex traits: guidelines for interpreting and reporting linkage results, Nature Genetics, 1995.

Thank you for this comment.

Indeed, it is attractive to envision that one component of the immune system (e.g. part of the innate or the adaptive immune system) would be a point of convergence across all neurodegenerative disorders. However, given the complexity of both the human immune system and the brain, we do not intend to describe convergence across all these disorders. Converging mechanisms are likely an over-simplification given the complexity. Rather, we present examples of individual results demonstrating potential immune mechanisms using the high-resolution data within the regional genetic correlation approach, which influence specific neurodegenerative diseases. Presenting individual results as we have done here is also of value. It promotes further research of particular loci in particular cell types; for instance, for functional experiments led by cell biologists and eventually for drug repurposing or development. We have been transparent on our approach and presentation of results, and we have attempted to find a balance between Reviewer’s 2 request of focusing on results related to particular diseases (Alzheimer’s and Parkinson’s) and this request for a broader outlook that does not highlight individual trait-gene pairs.

Indeed, we are striving to be transparent about what was tested and how our results were derived. To ensure transparency and reproducibility by the broader community, we have made all code available on GitHub, and in the text have explained all chosen filters and significance thresholds (and added a visualization of the filtering steps for further clarity, as suggested, Supplementary Figure 2). Although we highlight selected examples in the results to aid in readability, we do present all results tested on our publicly-available GitHub page, and now we have also added these tables to the Supplement for increased visibility (Supplementary Files 1-2). We have ensured that results are not over-stated, but rather our aim is that these findings will promote further work to test the role of the immune system, including follow-up studies on the particular sc-eQTL – GWAS trait and pQTL – GWAS trait findings, in neurodegenerative disease.

Specific comments:

The table of included studies could be presented in the main text. Given the sample size is important for all of the downstream analyses, it could help the reader to have these easily findable. Also, could the authors provide the LDSC intercept next to the $h^2(\text{obs})$

estimate?

We have now moved Supplementary Table 1 to the main text as Table 1, have renumbered the remaining Supplementary Tables accordingly, and we have added the LDSC intercept into the table.

It might help to add N=.... to figure 1 to show that LBD and FTD studies are relatively underpowered for these analyses.

Thank you. We have now added reference to Table 1 for sample sizes in the legend of Figure 1 to acknowledge differences in power among the included datasets.

To limit the multiple testing burden the authors limit the search space for regional correlations. For example, between disease traits only regions with at least 1 genome-wide significant signal and significant univariate regional genetic signal in both traits are included. This limits the number of test and leads to more lenient Bonferroni correction.

A similar filtering prior to correlation testing is applied for eQTL and pQTL.

The author could consider visualizing this in a flow chart so the reader can better interpret the exact filtering and reduction in number of tests in relation to all possible tests.

Thank you. As suggested, to make the filtering steps clearer, we have now included a visualization of the workflow displaying the filtering steps used for the the regional genetic correlations (across GWAS datasets, with sc-eQTLs and with pQTLs) as Supplementary Figure 2 (Figure pasted below for reference.)

Legend for Supplementary Figure 2: Flowchart of the number of tests included in each of the regional genetic correlation analyses.

The results section states: “we confirmed the expected significant positive correlations across these diseases, providing internal validation of the robustness of our approach”.

What do the authors mean by “robustness of our approach” as these are results from published data with widely used and validated tools using a standard approach. Indeed, most of these expected positive correlations have been published using the exact same data and methods.

We have removed the clause in this sentence on the robustness of our approach to avoid ambiguity: “Between immune-mediated diseases (our control traits), we confirmed the expected significant positive correlations across these diseases (Figure 1).”

The authors highlight some regional genetic correlations between neurodegenerative diseases and mention the CBR4, NEK1, CLCN3, C4orf27, SH3RF1 locus in ALS, CD and PD. This locus was not fine mapped in Parkinson’s disease (CLCN3 was only the nearest gene) and NEK1 is a known ALS risk gene in this locus, not CLCN3.

Thank you. We have now revised the sentence: “The gene *CLCN3* is the nearest gene at a disease risk locus for PD, whereas *NEK1* within this locus is a known ALS risk gene.”

Figure 3 displays the relative number eQTL correlations per cell type on y-axis. For a better comparison the authors could add a panel with the absolute number of eQTL correlations to show number is lower in AD, PD and ALS. For extra clarity the total number of genome-wide significant loci could be described as these are higher in SCZ compared to AD/PD/ALS.

Thanks for this suggestion. We have added the absolute number of significant eQTL correlations on Figure 3 for each cell type and the total per trait and added a brief description in the Results section: “Of note, the absolute number of significant correlations was not solely driven by the number of significant GWAS signals. For instance, SCZ is highly polygenic, but CD was the disease with the highest number of significant correlations.” Additionally, we have now tested whether the proportion of significant correlations per a particular trait-cell type combination is significantly different (i.e. not just qualitatively different) and added the results to the legend of Figure 3. See response to Reviewer 2’s third comment for details.

In line 299, what do the authors mean by a locus with a nominally significant p-value? Is this a SNP within this locus with $P < 0.05$? In that case, why is this relevant because (almost) every region has a SNP with $P < 0.05$ given the SNP density and multiple testing burden.

We apologize for the lack of clarity here. The p-value thresholds refer to whether or not the genetic correlations are unlikely to be seen by chance, rather than the individual SNP-trait associations. That is to say, we are referring to nominally significant genetic correlations ($p < 0.05$) between other pairs of tested neurodegenerative diseases.

References are not formatted uniformly. Journal name, edition, page etc are often missing. For example ref 18 (journal missing), 22 (name misspelled), 26 (journal

misspelled), 44 (pages missing). GWAS on ALS (van Rheenan et al) is cited twice (2016 and 2021)

Thank you for pointing out the various formatting issues in our reference list. We have updated the references in our reference manager and ensured that formatting is now consistent across references.

Reviewers' comments:

Reviewer #1 (Remarks to the Author):

The authors have responded satisfactorily to my comments.

Reviewer #2 (Remarks to the Author):

The authors carefully reviewed the manuscript and addresses my concerns. All my comments were addressed, and I don't have further comments.

Dr. Niccolo Tesi

Reviewer #3 (Remarks to the Author):

I appreciate the authors reasoning that highlighting genes and pathways related to the immune system in neurodegenerative diseases may help to guide future experiments. The choice for immune-cell-derived scRNA-seq is fair, based on literature and prior believes. This is not, however, a hypothesis free approach guided by the data at hand.

The authors argue that most of the sc-eQTL will be cell-type specific and therefore highlight the immune system. I think this assumption is not backed up by data or evidence. The argument that blood samples cluster away from other tissues in the 2015 GTEx paper, leaves out the fact that the same paper shows a high proportion of eQTL-sharing (fig 2 B-D) across tissues (probabilities > 0.55). Subsequent studies have confirmed the high correlation of eQTL between blood and brain tissues (correlation estimates between 0.75 - 0.8, <https://www.nature.com/articles/s41467-018-04558-1>). Furthermore, others have shown that there is no enrichment of tissue-*specific* eQTL for immune-mediated diseases and brain diseases compared to eQTL *shared* between blood and brain (<https://link.springer.com/article/10.1007/s10519-018-9914-2>).

I think it is important to acknowledge these assumptions and (arbitrary) choice to highlight the immune system. When you start from an unbiased angle and run enrichment analyses with the same GWAS and a similar toolset (LDSC or FUMA/FUMA-sc) across tissues and cell-types, not all neurodegenerative traits show enrichment for immune-mediated tissues and cell-types. This is clearly the case in AD (Schwartzentruber 2021), but not for PD (Nalls 2019) or ALS (Van Rheenen 2021). In PD and ALS there is enrichment for brain tissue in contrast to blood/lung/spleen in AD. Furthermore, sc-RNAseq enrichment shows enrichment for neurons in PD and ALS in contrast to microglia in AD. These results are presented in all three original papers, but not discussed by the authors.

I think this point should be raised in the discussion.

N.b. I think the authors have done a good job at making their analyses more transparent by adding supplementary figure 2.

We thank all reviewers and the editor for taking the time to assess the revised version of our manuscript. Below we outline (in bold) how we have modified the manuscript to address the remaining suggestions from Reviewer 3. We describe how we have updated the discussion part of the manuscript to include previous studies to discuss and strengthen the reason behind our focus on the immune system. We believe that this feedback has improved our manuscript, and we look forward to your response.

Response to reviewers' comments:

Reviewer #3

I appreciate the authors reasoning that highlighting genes and pathways related to the immune system in neurodegenerative diseases may help to guide future experiments. The choice for immune-cell-derived scRNA-seq is fair, based on literature and prior believes. This is not, however, a hypothesis free approach guided by the data at hand.

The authors argue that most of the sc-eQTL will be cell-type specific and therefore highlight the immune system. I think this assumption is not backed up by data or evidence. The argument that blood samples cluster away from other tissues in the 2015 GTEx paper, leaves out the fact that the same paper shows a high proportion of eQTL-sharing (fig 2 B-D) across tissues (probabilities > 0.55). Subsequent studies have confirmed the high correlation of eQTL between blood and brain tissues (correlation estimates between 0.75 - 0.8, <https://www.nature.com/articles/s41467-018-04558-1>). Furthermore, others have shown that there is no enrichment of tissue-*specific* eQTL for immune-mediated diseases and brain diseases compared to eQTL *shared* between blood and brain (<https://link.springer.com/article/10.1007/s10519-018-9914-2>).

I think it is important to acknowledge these assumptions and (arbitrary) choice to highlight the immune system. When you start from an unbiased angle and run enrichment analyses with the same GWAS and a similar toolset (LDSC or FUMA/FUMA-sc) across tissues and cell-types, not all neurodegenerative traits show enrichment for immune-mediated tissues and cell-types. This is clearly the case in AD (Schwartzentruber 2021), but not for PD (Nalls 2019) or ALS (Van Rheenen 2021). In PD and ALS there is enrichment for brain tissue in contrast to blood/lung/spleen in AD. Furthermore, sc-RNAseq enrichment shows enrichment for neurons in PD and ALS in contrast to microglia in AD. These results are presented in all three original papers, but not discussed by the authors.

I think this point should be raised in the discussion.

N.b. I think the authors have done a good job at making their analyses more transparent by adding supplementary figure 2.

We thank the reviewer for their careful review of our revised manuscript and for this feedback. We acknowledge the comments and evidence regarding the shared eQTLs. Our analysis is not hypothesis-free, and in this revised version of the manuscript, we have acknowledged this choice more clearly in the discussion section. We note that pinpointing targets in specific cell types involved in the immune system can aid in downstream repurposing of existing immune therapies as treatment options for certain neurodegenerative diseases. Certainly, the magnitude of the immune contribution is likely stronger in certain neurodegenerative disorders (e.g. AD) and weaker in others. There, nevertheless, is value in assessing the immune component, even if the signal is weaker, across neurodegenerative disorders given the actionability inherent to immune-mediated targets, i.e. for drug repurposing, as opposed to targets within the brain, for example. Given the points raised, we have now revised the discussion part of the manuscript to highlight previous studies and strengthen the reason behind our choice to focus on the immune system. Specifically:

Here, we chose to focus on the role of the immune system in neurodegeneration. Indeed, for certain neurodegenerative diseases, the role of the immune system is arguably more prominent than for others. For AD, for instance, there is substantial evidence of the role of the immune system, with findings of immune-mediated tissues being enrichment for heritability and of single-cell RNA-sequencing enrichment analyses pointing to microglia (Jansen et al. 2019, Schwartzentruber et al. 2021). Work has also highlighted activated microglia and T cell responses in tauopathies (Chen et al. 2023). Studies of other neurodegenerative diseases have identified the HLA region on chromosome 6 as being implicated in disease risk; for example, the HLA region is a genome-wide signal in PD GWASes (Nalls et al. 2014, Nalls et al. 2019), and there is evidence of immune-mediated genetic enrichment for FTD within this region (Broce et al. 2018). Furthermore, for PD, there is recent converging evidence of immune-related influences on disease risk through the role of T cells in brain inflammation and neurodegeneration (Dhanwani et al. 2022, Williams et al. 2021). Altogether, these lines of evidence, in combination with the utility of repurposing existing immune-based treatments, motivated our focus on immune-mediated factors presented here. Nevertheless, we acknowledge evidence of enrichment in non-immune cell and tissue types across neurodegenerative diseases. For instance, for PD and ALS, analyses have demonstrated heritability enrichment for brain tissue, and work based on single-cell RNA-sequencing has shown enrichment for neurons for these two diseases (Nalls et al. 2019, Van Rheenen et al. 2021). Likely, neurodegenerative disease risk and progression are mediated by a complex interplay of multiple cell and tissue types influencing a diverse set of biological systems. Neurodegeneration is not limited to a single system such as the central nervous system or, as assessed here, the immune system. Indeed, future research is warranted to further investigate immune-mediated mechanisms on neurodegenerative disease risk, and to expand explorations to assess other biological systems to improve our understanding of additional disease mechanisms.

REVIEWERS' COMMENTS:

Reviewer #3 (Remarks to the Author):

I would like to thank the authors for their response and revising the discussion. They provide a concise context in which a broader audience can appreciate their valuable work.